# Human Preadipocytes Differentiated under Hypoxia following PCB126 Exposure during Proliferation: Effects on Differentiation, Glucose Uptake and Adipokine Profile

**DOI:** 10.3390/cells12182326

**Published:** 2023-09-21

**Authors:** Zeinab El Amine, Jean-François Mauger, Pascal Imbeault

**Affiliations:** 1School of Human Kinetics, Faculty of Health Sciences, University of Ottawa, Ottawa, ON K1N 6N5, Canada; zelam100@uottawa.ca (Z.E.A.); jmauger@uottawa.ca (J.-F.M.); 2Institut du savoir Montfort, Hôpital Montfort, Ottawa, ON K1K 0T2, Canada

**Keywords:** hypoxia, polychlorinated biphenyl, inflammatory adipokines, human differentiated adipocytes, AhR, ARNT

## Abstract

Persistent organic pollutants (POPs) accumulation and hypoxia are two factors proposed to adversely alter adipose tissue (AT) functions in the context of excess adiposity. Studies have shown that preadipocytes exposure to dioxin and dioxin-like POPs have the greatest deleterious impact on rodent and immortalized human preadipocyte differentiation, but evidence on human preadipocytes is lacking. Additionally, hypoxia is known to strongly interfere with the dioxin-response pathway. Therefore, we tested the effects of pre-differentiation polychlorinated biphenyl (PCB)126 exposure at 10 µM for 3 days and subsequent differentiation under hypoxia on human subcutaneous adipocytes (hSA) differentiation, glucose uptake and expression of selected metabolism- and inflammation-related genes. Pre-differentiation PCB126 exposure lowered the adenosine triphosphate (ATP) content, glucose uptake and leptin expression of mature adipocytes but had limited effects on differentiation under normoxia (21% O_2_). Under hypoxia (3% O_2_), preadipocytes ability to differentiate was significantly reduced as reflected by significant decreased lipid accumulation and downregulation of key adipocyte genes such as peroxisome proliferator-activated receptor gamma (PPARγ) and adiponectin. Hypoxia increased glucose uptake and glucose transporter 1 (GLUT1) expression but abolished the adipocytes insulin response and GLUT4 expression. The expression of pro-inflammatory adipokine interleukin-6 (IL-6) was slightly increased by both PCB126 and hypoxia, while IL-8 expression was significantly increased only following the PCB126-hypoxia sequence. These observations suggest that PCB126 does not affect human preadipocyte differentiation, but does affect the subsequent adipocytes population, as reflected by lower ATP levels and absolute glucose uptake. On the other hand, PCB126 and hypoxia exert additive effects on AT inflammation, an important player in the development of chronic diseases such as type 2 diabetes and cardiovascular diseases.

## 1. Introduction

Proper adipose tissue (AT) maintenance through proliferation and differentiation of preadipocytes is essential for whole-body energy homeostasis [1,2]. The accumulation of lipophilic persistent organic pollutants (POPs) and hypoxia are two factors that are recognized to adversely alter AT metabolism and to contribute to the development of obesity-related metabolic complications [3,4,5].

POPs are synthetic compounds that consist of many groups of halogenated compounds, such as polychlorinated biphenyls (PCBs). POPs, which bioaccumulate in the food chain and resist biodegradation, are commonly referred to as “metabolic disruptors” [6,7]. Traditionally, 2,3,7,8-tetrachlorodibenzo-*p*-dioxin (TCDD) and dioxin-like PCBs have been reported to exhibit their toxicity by activating the aryl-hydrocarbon receptor (AhR), a transcription factor that regulates the expression of genes containing a xenobiotic response element (XRE) in their promoter [8]. TCDD and dioxin-like PCBs have been shown to alter the acquisition of adipocyte phenotype (also called “adipocyte differentiation”) and adipocyte metabolism [9,10]. For instance, it has been reported that 3T3-L1 cells and immortalized human preadipocytes (Normal Pre-Adipocytes, NPADs) exposure to dioxin and dioxin-like PCB126 prior to differentiation impaired subsequent adipocyte differentiation [11,12,13]. In NPADS, the impairment of adipocyte differentiation by PCB126 has later been associated with inflammation, specifically the activation of the nuclear factor kappa-light-chain-enhancer of activated B cells activation (NFkB) [14].

Hypoxia, another important metabolic disruptor, is known to profoundly affect adipocyte functions [15,16]. For instance, acute exposure (24 h) to low oxygen tension is well known to decrease peroxisome proliferator-activated receptor γ (PPARγ) expression and to increase leptin secretion in differentiated human adipocytes [17,18,19,20]. Acute hypoxia exposure also impacts the adipocyte’s lipid storage and mobilization functions by reducing the activity of the lipoprotein lipase activity while increasing basal intracellular lipolysis [21,22]. Moreover, exposure to acute hypoxia have been shown to increase GLUT1 facilitative glucose transporter gene and protein levels, glucose transport [19,23] as well as expression and release of inflammation-related adipokines [17,24,25] in human differentiated adipocytes. These cellular responses to hypoxia are mostly exerted via the genomic action of the hypoxia inducible factor 1a (HIF-1α), a transcription factor that is stabilized and activated when O_2_ levels drop below 4–5% [26,27,28,29].

Interestingly, hypoxia and dioxins signaling both exploit the aryl hydrocarbon receptor nuclear translocator (ARNT) transcription factor, which can be recruited by both AhR and HIF-1α thus establishing a significant foundation for a possible crosstalk between these two signaling pathways [4,30]. We recently showed in differentiated human adipocytes that 24 h of hypoxia and PCB126 co-exposure resulted in an increase in vascular endothelial growth factor-A (VEGFA) expression, a traditional marker of HIF-1α activation, and in a strong decrease in the transcription of CYP1A1, a detoxifying enzyme member of the cytochrome P450 superfamily and a traditional marker of AhR activation. [17]. In the same study, we reported that exposure to PCB126 or hypoxia alone or in combination increases IL-6 secretion in a dose dependent manner, which suggests that long-term co-exposure of adipocytes to hypoxia and POPs could favor the establishment of a pro-inflammatory state in the AT.

Considering the previously reported effects of dioxin-like PCBs and hypoxia on adipocytes, we exposed proliferating human preadipocytes to PCB126 and then differentiated them under hypoxia to determine how the combination of both stressors would affect adipocyte differentiation, glucose uptake and key inflammatory markers.

## 2. Methods

### 2.1. Cell Culture

Pooled human subcutaneous preadipocytes from donors with a body mass index over 30 kg/m^2^ (SP-F-3) were obtained from Zenbio (Durham, NC, USA) and cultured according to the manufacturer’s instructions. Cells suspended in Zenbio’s PM-1 culture media were seeded in culture-treated 96-well plates at a density of at least 40,000 cells/cm^2^. Cells were then placed in a HeraCell 150iO_2_ incubator (Thermo Fisher Scientific, Waltham, MA, USA) incubator maintained at 100% humidity and 5% CO_2_. After 3 days of proliferation, medium was replaced by Zenbio DM-2 to induce differentiation. After 7 days of differentiation, the medium was partially replaced with Zenbio AM-1 in which cells remained until day 14 after induction.

### 2.2. PCB126 and Hypoxia Exposure

During proliferation, which was carried for 3 days, preadipocytes were exposed to either 10 µM PCB126 in dimethyl sulfoxide (DMSO) or DMSO alone (final DMSO concentration 0.1% in both conditions). Media were not changed during the proliferation phase. A concentration of 10 μM PCB126 was used because it has the greatest detrimental effect on the differentiation capacity of NPAD cells without inducing selective cytotoxicity [12] and it is well tolerated by differentiated human preadipocytes [17]. After 3 days of proliferation, media were switched to DM-2 containing no DMSO nor PCB126 to induce differentiation. At the same time, hypoxia exposure was initiated and carried out for 14 days in a HeraCell 150iO_2_ incubator (Thermo Fisher Scientific, Waltham, MA, USA) maintained at 3% O_2_ with medical nitrogen and 5% CO_2_. During the differentiation phase, media were partially changed every other day to avoid acidification of the culture media by lactate accumulation under hypoxia. This procedure was also carried out for cells cultured under normoxia.

### 2.3. Cell Expansion and Viability

Cell viability was assessed daily by visual examination using a Zeiss Axiovert 40 °C light microscope (Carl Zeiss, Göttingen, Germany) to detect anomalies such as cell detachment from culture ware. Twenty-four hours and 14 days following induction, cells expansion/viability was assessed by bioluminescent ATP quantification using CellTiterGlo 2 reagent from Promega (Madison, WI, USA). Because ATP content was affected by PCB126 and hypoxia, ATP measurements were refined by measuring the ADP/ATP ratio using the MAK135 assay kit from Millipore-Sigma (Oakville, ON, Canada). Lactate dehydrogenase (LDH) activity, another indicator of cell apoptosis and necrosis, was measured in culture media after 14 days of differentiation using the CyQUANT LDH cytotoxicity assay (Life Technologies Inc., Burlington, ON, Canada).

### 2.4. Glucose Uptake

Glucose uptake was measured using Promega bioluminescent GlucoseUptakeGlo assay based on 2-deoxyglucose (2-DG) uptake according to manufacturer instructions. Cells were first glucose-starved by washing 3 times with Zenbio basal medium (BM-1) containing no glucose followed by incubation for 1.5 h in glucose-free BM-1. Insulin was added to relevant wells to a final concentration of 100 nM 30 min before glucose uptake assessment. BM-1 was removed from wells and cells were incubated 12 min with 1 mM 2-DG in PBS. The GLUT-specific inhibitor Glutor [31] was used at a final concentration of 1 µM to confirm GLUT-specific glucose uptake. On average, Glutor decreased glucose uptake by 90% under normoxia and by 85% under hypoxia. Glucose uptake was normalized to cellular ATP to account for the possible effects of treatments on cell expansion and/or energy metabolism. ATP measurements considered for this adjustment came from Glutor wells because 2-DG uptake is known to deplete cellular ATP as 2-DG is phosphorylated but not further metabolized [32].

### 2.5. TG Content

TG concentrations were measured in glucose uptake cell lysate using the Wako L-type Triglyceride M (FUJIFILM Wako Chemicals U.S.A. Corporation, Richmond, VA, USA) assay adapted for microplates.

### 2.6. Gene Expression

Cells were rapidly washed 3 times with ice-cold PBS and lysed using Qiagen RLT buffer containing 1% Beta-mercaptoethanol. Lysates were spun on Qiagen QiaShredder columns and lysates were kept at −80 °C until further processed. Total ARN was extracted using Qiagen RNeasy mini kit and concentrated using Qiagen MinElute clean up kit when necessary. Complementary DNA (cDNA) was obtained using the Qiagen Quantitect Reverse transcription kit and the maximum volume of RNA template allowed by the kit (12 µL) on a T-personal Combi thermocycler (Biometra, Gottingen, Germany) and real-time quantitative polymerase chain reaction (RT-qPCR) was conducted using Quantitect primers (Qiagen, Germantown, MD, USA) and MBI EVOlution EvaGreen rtPCR mix (Montreal Biotech Inc., Dorval, QC, Canada) on a RG-3000 Rotor-Gene (Corbett Research Ltd., Mortlake, Australia). The Quantitect primers used had the following Qiagen catalog numbers: Adiponectin QT00014091; Beta-actin QT00095431; CYP1A1 QT00012341; GLUT1 QT00068957; GLUT4 QT00097902; IL-6 QT00083720; IL-8 QT00000322; Leptin QT00030261; PPAR-gamma QT00029841; VEGFA QT01682072. Melting curve analyses were performed to ensure that amplification yielded single products. Amplification curves were analyzed using the Rotor-Gene Q Series software version 2.2.3. Amplification efficiency was determined for each individual amplification reaction using the comparative quantitation feature of the Q software, which is based on the fluorescence increase over 4 cycles following “take-off” of the fluorescence signal. The “take-off” point corresponds to the point at which the second derivative of the amplification curve reaches 20% of its maximum (determined automatically by the Rotor-Gene Q software). Average amplification factors were very stable within and across genes (averaging 1.7 to 1.8). Fluorescence thresholds for *CT* determination were determined for each gene in the first experiments and arbitrarily assigned the value corresponding to half the average fluorescence at take-off cycle. Fold-changes were calculated using the formula [33]:$$Fold\;change=\frac{E_{tar}^{\;\;\left(CT_{tar\;ctl}-CT_{tar\;treat}\right)}}{E_{ref}^{\;\;\left(CT_{ref\;ctl}-CT_{ref\;treat}\right)}}$$
*where E corresponds to the average amplification efficiency of each gene within the same experiment (i.e., E was averaged from individual reactions for each gene for each experiment).*


### 2.7. Statistical Analyses

Data were analyzed by linear mixed modeling using restricted maximum likelihood with “proliferation” (PCB126 or DMSO during proliferation) and “differentiation” (21% or 3% O_2_ during differentiation) as main effects and “experiment” as cluster variable. Intercept was set as random across experiments to account for correlation between measurements within individual experiment. Three independent experiments were conducted, and each combination of treatment was tested in duplicate (2 wells) within each experiment. We included individual data from replicate wells in the statistical design because replicate wells per experiment were minimal and because we believe within-experiment replicates provide valuable information on the robustness of the biological effect. For clarity’s sake, purely technical replicates (i.e., duplicate measurements from a same well for a given variable) were averaged and contributed unique datum to the statistical model. For gene expression, statistical analyses were conducted on ΔΔCT values and for glucose uptake and triglyceride content, analyses were conducted on raw luminescence values or ratios of raw luminescence values. Significant interactions were further investigated by post hoc multiple comparison analyses using Bonferroni correction. Figures represent main effects as averaged percent of control condition (proliferation in DMSO and differentiation under normoxia) within each experiment to better illustrate treatment effects by excluding part of the inter-experiment variability. Analyses were performed using the Jamovi statistical software version 2.2.5 with the GAMJL module version 2.6.6.

## 3. Results

### 3.1. Effects of Treatments on Cell Viability and Growth

The daily visual examination of the cells showed no obvious anomaly such as reduced cell proliferation or cell detachment from culture ware in response to treatments throughout the experiment duration (17 days total) (Figure 1). After 72 h of proliferation, cellular ATP was slightly lower (−7%, *p* = 0.009) in PCB126-exposed cells but the ADP/ATP ratio and media lactate dehydrogenase (LDH) activity were similar between PCB126- and DMSO-exposed cells (*p* = 0.981 and 0.248), suggesting that PCB126 had only a minor effect on preadipocytes proliferation. After 24 h of differentiation, ATP content was no more affected by PCB126 exposure while a 20% decrease in ATP (*p* = 0.003) (Figure 2) and a minor increase in ADP/ATP ratio (0.247 vs. 0.224, *p* = 0.004) were observed in cells exposed to hypoxia. LDH activity in the media was not performed at that time. After 14 days of differentiation, both PCB126 and hypoxia treatments were associated with a significant ~30% decrease in ATP content (main effect of PCB126 *p* = 0.014, main effect of hypoxia *p* = 0.02) (Figure 2). Media LDH activity was not different between cells exposed to PCB126 or DMSO at that time, but media LDH activity (Figure 2) and ADP/ATP ratio were, on average, reduced by ~50% in cells exposed to hypoxia (main effect of hypoxia *p* < 0.001 for both variables).

### 3.2. Gene Expression

VEGFA expression was measured as an indicator of the induction of HIF signaling by hypoxia. VEGFA expression was increased two-fold after 14 days of hypoxia (main effect of hypoxia *p* < 0.001) (Figure 3) and this effect was not altered by PCB126 exposure during proliferation. In contrast, there was a significant PCB126 × hypoxia interaction (*p* < 0.001) for CYP1A1 expression, a gene reflecting the induction of AhR signalling by PCB126. Specifically, following the initial PCB126 exposure, CYP1A1 expression was significantly increased after 14 days of PCB126-free normoxia or hypoxia treatment, this effect being greater upon normoxia (100-fold vs. 3-fold) (Figure 3).

Adipose specific genes were affected differently by hypoxia and PCB126 pre-exposure. PPARγ and adiponectin expression were significantly suppressed under hypoxia by 5- and 100-fold, respectively (main effect of hypoxia *p* < 0.001 in both cases) (Figure 4). PPARγ and adiponectin transcript levels were unaffected by PCB126 pre-exposure. In contrast, leptin expression was significantly increased by hypoxia, but was lowered by pre-differentiation PCB126 exposure. The leptin-lowering effect of PCB126 was significantly attenuated by hypoxia (PCB126 x hypoxia interaction *p* = 0.003) (Figure 4).

The expression of pro-inflammatory adipokine IL-6 was slightly but significantly increased by both PCB126 exposure during proliferation and hypoxia exposure during differentiation (*p* = 0.009 and *p* = 0.006, respectively) (Figure 5). IL-8 expression, on the other hand, was significantly increased by PCB126 exposure only in cells subsequently exposed to hypoxia during differentiation (PCB126 × hypoxia interaction *p* = 0.013) (Figure 5).

### 3.3. TG Content and Glucose Uptake

Adipocyte TG content decreased by more than 80% following 14 days of hypoxia exposure (main effect of hypoxia *p* < 0.001) (Figure 6). Interestingly, when TG content was normalized to ATP content, hypoxia was still associated with a reduction in TG content, but PCB126 exposure during proliferation tended to be associated with greater TG content, especially under normoxia (PCB126 × hypoxia interaction *p* = 0.06) (Figure 6).

After 24 h of differentiation, basal glucose uptake (either unadjusted or adjusted for ATP content) was increased by ~50% under hypoxia (main effect of hypoxia *p* < 0.001), regardless of PCB126 exposure during proliferation (Figure 7). After 14 days of differentiation, basal glucose uptake unadjusted for ATP was still increased in response to hypoxia but was significantly reduced in cells exposed to PCB126 during proliferation (main effects of hypoxia and PCB126 both *p* < 0.001) (Figure 8). However, when glucose uptake was adjusted for cells ATP content, only the effect of hypoxia remained significant (main effect of hypoxia *p* < 0.001, main effect of PCB126 *p* = 0.610) (Figure 8). Glucose uptake stimulation by insulin was only assessed at day 14 and was virtually completely abolished by differentiation under hypoxia while being unaffected by PCB126 exposure during proliferation (main effect of hypoxia *p* < 0.001) (Figure 9). The expression of glucose transporters GLUT1, the main transporter responsible for basal glucose uptake, and GLUT4, responsible for the increase in glucose transport in response to insulin, were consistent with the ATP-adjusted glucose uptake rates. PCB126 had no effect on either GLUT1 or GLUT4 expression, while hypoxia induced a two-fold increase in GLUT1 expression and virtually abolished GLUT4 expression (main effect of hypoxia for both gene *p* < 0.001) (Figure 10).

## 4. Discussion

A growing body of evidence shows that hypoxia and POPs accumulation in AT, two features observed when AT mass excessively expands, cause endocrine and metabolic alterations [6,7,9,10,15,16]. Several studies suggested that AhR activation through exposure to chemical insults such as TCDD or PCBs induces inflammation in AT [34,35,36,37], and impairs adipocyte differentiation [7,38,39]. However, these studies were conducted almost exclusively in guineapigs and rodents. Also, when the effects of organic pollutants are assessed in vitro, exposure is almost always conducted throughout or at the end of the differentiation phase. However, in the case of the murine 3T3-L1 adipogenic line, the inhibition of adipocyte differentiation occurs exclusively when cells are exposed to TCDD before induction of differentiation [11,13]. Work on a recently developed human adipogenic immortalized cell line (NPAD cells) indicated that PCB126 exposure during proliferation also impairs differentiation in these adipocyte precursors [12]. Nonetheless, this effect was not easily reproduced in human mesenchymal stem cells using standard culture conditions [40]. Since the effect of early dioxin-like organic pollutants exposure on the differentiation capacity of adipocyte precursor seems highly variable among cell lines, we investigated whether PCB126 exposure prevents the differentiation of human subcutaneous preadipocytes. Given our recent observation indicating that hypoxia strongly inhibits dioxin signaling, we also sought to examine how chronic hypoxia following early PCB126 exposure could affect adipocyte differentiation. Our results show that pre-differentiation exposure to PCB126 had no significant effect on adipocyte differentiation and glucose uptake in hSA. Instead, after 14 days of differentiation, the main effect of early PCB126 exposure was a minor decrease in cellular ATP content and significant modulation in the expression of adipokines and adipose specific genes. Conversely, differentiation under continuous hypoxia strongly suppressed adipocyte differentiation, profoundly altered the expression of adipokines and adipose-specific genes and exacerbated the PCB126-induced inflammatory response.

### 4.1. Effects of PCB126 Exposure on Human Preadipocytes under Normoxia

CYP1A1 is a membrane-bounded hemoproteins part of the cytochrome P450 (CYP) enzyme family that plays a key role in the detoxification of xenobiotics [41,42]. CYP1A1 induction is mostly transcriptional, and this mainly occurs by the AhR mechanism [41,42]. We and others have previously shown that CYP1A1 expression was significantly induced by PCBs exposure in hSA [12,17,43]. In the current study, we found that CYP1A1 expression is still significantly induced (100-fold) in mature adipocyte obtained from preadipocytes that were exposed to PCB126 14 days earlier. The persistence of high CYP1A1 transcript levels in mature adipocytes following PCB126 exposure during preadipocyte proliferation has already been reported in NPAD cells [12]. Together, the induction of CYP1A1 expression indicates that human preadipocytes are sensitive to xenobiotics compounds that activate the AhR signalling pathway.

In the present study, pre-differentiation exposure to PCB126 had no obvious consequence on adipocyte differentiation under normoxia, as demonstrated by the absence of alteration in PPARγ and adiponectin expressions as well as in TG content. This finding differs from the previous study, showing that exposure of NPAD cells to PCB126 resulted in significant reduction in their ability to fully differentiate into adipocytes [12]. This divergence could be attributed to several factors. First, in both studies, cells were exposed to PCB126 until confluence was reached, but this amounted to 3 days in the present study, compared to 6 days in the study of Gadupudi et al. [12]. Since PCBs have been shown to have different long term deleterious effects, such as DNA adducts formation, a longer incubation in the presence of PCB126 may have altered the cell cycle and the ability of NPAD cells to differentiate [44,45]. Second, the examination of the data reported by Gadupudi et al. suggests that NPAD cells seem to proliferate in a significant manner after confluence is reached [12]. NPAD cells could, therefore, be resistant to enter quiescence, which would be consistent with the seemingly low differentiation rates (25% in control conditions) that are concomitantly reported [12]. This, in turn, suggests that NPAD cells may be natively less efficient at differentiating into adipocyte, making them, perhaps, more sensitive to disrupting factors such as dioxin-like PCBs. Also, a more recent report by the same research group shows that treating human adipose mesenchymal/stromal cells in a similar fashion with Aroclor, a POP mixture containing PCB126 among others, does not impair their differentiation into adipocytes unless fetal bovine serum is almost completely removed from the culture media [40]. This latter observation further supports the fact that different cell lines respond differently to POP exposure, which makes our observation that human subcutaneous preadipocytes behave differently from NPAD or 3T3-L1 cells less surprising.

Importantly, the inhibitory effect of PCB126 on NPAD differentiation reported by Gourronc et al. was accompanied by an inflammatory response characterized by a strong upregulation of IL-8 expression [14]. We and others showed previously that acute exposure to PCB126 increases the production of IL-8 in differentiated human adipocytes [17] and human multipotent adipose-derived stem cells [36]. However, in the present study, the PCB126-normoxia sequence did not affect IL-8 expression, but instead led to a slight but significant increase in IL-6 expression, as previously observed in human monocytes or endothelial cells by others [45,46]. While it cannot be excluded that IL-8 expression may have been transiently increased at any time point during proliferation or differentiation, this eventuality had no apparent impact on the capacity of human subcutaneous preadipocyte to differentiate into adipocytes under normoxic conditions.

Interestingly, pre-differentiation exposure to PCB126 significantly decreased leptin expression under normoxia. In our previous work [17], differentiated adipocytes acutely exposed to PCB126 for 48 h had no significant change in leptin secretion. Lower serum leptin levels were reported in the Yusho victims highly exposed to PCBs and polychlorinated dibenzofurans (PCDF) [47], but this effect was not observed in rats exposed to 50 g/kg of TCDD [48]. How dioxin-like pollutants affect leptin secretion is currently poorly understood. Nonetheless, the current study provides preliminary evidence that human preadipocyte exposed to PCB126 appears to yield mature adipocytes that produce less leptin. Given the strong and positive association with leptin levels and body adiposity [49,50], whether dioxin exposure during AT development could, in the long term, alter leptin levels and modulate the regulation of energy balance in humans remains to be explored.

Basal glucose uptake unadjusted for ATP was not affected by PCB126 after 24 h of differentiation but was significantly lower after 14 days under normoxia. The absolute glucose uptake under insulin stimulation was also lower in cells that were exposed to PCB126); however, when expressed as fold increase relative to basal glucose uptake, the insulin response appeared not affected by PCB126 exposure. In vivo and in vitro studies have linked dioxin and dioxin-like PCBs to impaired glucose uptake and glucose intolerance, but these studies were all conducted in guineapigs, rodents or rodent cell lines [35,39,51,52], which may behave differently in response to dioxin and other organic pollutants. The present study, therefore, suggests that preadipocyte exposure to PCB126 may produce populations of mature adipocytes displaying global reduced glucose uptake.

It is important to note that cells’ ATP content, although initially unaltered in cells exposed to PCB126, was significantly reduced 14 days post exposure. Interestingly, when glucose uptake on day 14 is normalized to ATP content, the reduction in glucose uptake induced by PCB126 disappears. The reason behind the decrease in cellular ATP is not obvious since neither media LDH activity nor ADP/ATP ratio provide evidence of decreased cell viability in response to PCB126 exposure. It is, however, interesting to note that ATP correction makes glucose uptake measurements fall in line with the observation that GLUT1 and GLUT4 expressions (the main basal and insulin-stimulated glucose transporters) are not altered by PCB126. We believe that the concordance of these two relative measurements (glucose uptake relative to ATP and GLUTs relative to B-actin) is not trivial and, combined with the rest of our observations, suggest that human preadipocytes exposed to PCB126 produce normally differentiating adipocytes, but with a limited capacity for glucose uptake that could be attributable to lower ATP levels. Our study unfortunately provides no explanation for the delayed decrease in cellular ATP following PCB126 exposure other than that it does not appear to be apoptosis- or necrosis-related, based on media LDH activity and the ADP/ATP ratio. Dioxin-like pollutants have been shown to produce DNA adducts in the long term, which can lead to cell cycle arrest [53]. It is, therefore, possible that PCB126/Ahr may induce slight impairments in human preadipocyte proliferation that may not produce noticeable effects after a few doubling times (i.e., 4 days) but may become significant after several (i.e., after 17 days) [12]. However, the visual examination of the cultures (Figure 1) did not support a substantial difference in cell number between cultures initially exposed to DMSO and those exposed to PCB126. Another possible explanation for the decrease in ATP levels observed in cells exposed to PCB126 before differentiation could reside in the fact that POPs have been shown to impair mitochondrial oxygen consumption and ATP production, especially in the longer term, in vivo and in vitro [54,55,56]. While further investigations are required to explain the observed delayed decrease in ATP levels 14 days following PCB126 exposure, media LDH activity, cellular ADP/ATP ratio and visual examination strongly suggest that the cells’ viability was not compromised. The apparent relationship between the depleted ATP content and glucose uptake warrants further investigations.

### 4.2. Effects of Hypoxia on Differentiating Human Preadipocytes

The chronic exposure to hypoxia during differentiation was used to simulate the environment of hypoxic patches that may develop during the excessive development of AT mass [15]. VEGFA expression was increased significantly in adipocytes differentiated under hypoxia, indicating activation of the hypoxia response pathway persisting after 14 days. Cellular ATP levels were lower after both 24 h and 14 days of hypoxia. However, both media LDH activity and the ADP/ATP ratio were also lower after 14 days of hypoxia, suggesting that depressed ATP levels may not reflect increased apoptosis or necrosis, but could instead indicate a state of cellular hypometabolism and alterations in adenosine metabolism that has been proposed to occur under hypoxia [57]. A rapid decrease in cellular ATP unrelated to increase in ADP or cellular apoptosis/necrosis has also been reported in primary culture rat hepatocytes exposed to hypoxia [58]. Such a mechanism could be critical for cell survival under hypoxia by preventing oxidative stress that could arise from ADP-stimulation of oxidative phosphorylation under hypoxic conditions. To the best of our knowledge, this effect of hypoxia on adipocyte ATP content has not been reported before, and further studies will be required to better understand the mechanisms underlying this effect and how it may impact adipocyte function.

In the present study, hypoxia caused a clear visual reduction in adipocyte TG droplets size. This is consistent with our previous observation on human preadipocytes differentiated under chronic severe hypoxia and could be explained by a strong reduction in the expression of key adipogenic genes such as carbohydrate response element binding protein (ChREBP), acetyl-CoA carboxylase (ACC), fatty acid synthase (FASN), diacylglycerol acyl transferase 1 and 2 (DGAT1 and DGAT2) [21]. In the present study, the reduction in lipid droplets was accompanied by a strong repression of PPARγ expression, indicating that continuous hypoxia greatly represses adipocytes differentiation. These results are in line with other studies that utilized prolonged hypoxia exposure after differentiation induction [29,59]. The decrease in adiponectin expression and increased in leptin and Il-6 expression under hypoxia are also consistent with previous observations [17,60]. These alterations in lipid accumulation and adipokine profile further confirm that chronic severe hypoxia results in an impairment of differentiation in cultured human adipocytes.

Glucose uptake also behaved as expected in response to hypoxia exposure. After both 24 h and 14 days of exposure, a strong increase in basal glucose uptake was observed, either adjusted or not for ATP content concomitant to an increase in GLUT1 expression, as observed by Wood et al. [23]. However, the fact that glucose uptake remained significantly increased under hypoxia following ATP-adjustment is interesting and may reflect a shift in substrate partitioning toward glucose utilization and greater contribution of anaerobic glycolysis to energy metabolism under hypoxia. Finally, also as expected, hypoxia completely prevented insulin from promoting glucose uptake in adipocyte, which was accompanied by a strong repression of GLUT4 expression, as previously observed [18].

### 4.3. Interaction between Hypoxia and PCB126 on Differentiating Preadipocytes

We previously showed that hypoxia represses but does not completely abolish the induction of the dioxin response in human differentiated adipocytes acutely exposed to PCB126 [17]. Consistently, in the present study, most of the effects of PCB126 exposure, notably the decrease in leptin expression, were similarly obscured by 14 days of hypoxia. However, much like in our previous report, the induction of CYP1A1 by early exposure to PCB126 was strongly attenuated but still significantly present after 14 days of hypoxia. Together, these results confirm that hypoxia strongly interferes with, but does not completely abolish, the xenobiotic sensing pathway in adipocytes so that the combination of both factors could lead to additive effects.As such, IL-8 expression was not increased by PCB126 exposure followed by normoxia but was significantly increased following the PCB126-hypoxia sequence (PCB126 × hypoxia *p* = 0.013). PCB126 is increasingly well known to induce IL-8 expression in the short term (hours to days) in adipocyte cell lines, although this effect could be only transient [12,14]. Hypoxia, on the other hand, has been shown to acutely induce IL-8 expression in several cell types in culture (lungs, ovarian cancer, macrophages) but, to the best of our knowledge, this has not been reported in human or rodent adipocytes. The present study seems to confirm that, even in the long term, hypoxia does not induce IL-8 expression in hSA. Therefore, a likely explanation for the increase in IL-8 expression by the PCB126-hypoxia combination is that continuous hypoxia may have allowed PCB126 to persistently stimulate IL-8 expression after several days of PCB-free culture. One way hypoxia could have achieved this is by preventing adipogenesis and impairing the “safe” deposit of remaining lipophilic PCB126 in lipid droplets, the major sink for adipose dioxin since adipocyte do not express CYP1A2, which is the protein responsible for the preferential sequestration of dioxins in organs such as the liver. Another contributing factor could be that remaining PCB126 molecules may have been more slowly metabolized and detoxified due to the strong repression of CYP1A1 expression by hypoxia. However, experimental data demonstrated that human cytochrome P450 does not metabolize PCB126 [61], which, if accurate, would make this later explanation unlikely. Regardless, while the interaction between acute and long-term hypoxia requires confirmation and the mechanisms responsible for this interaction need to be elucidated, the present observation suggest that hypoxia could potentiate the inflammatory response induce by dioxin-like pollutants. This is especially relevant knowing that (1) humans are exposed to dioxin-like pollutants at early age (breast milk), (2) that the proportion of individuals living with obesity, a condition increasingly recognized as inducing AT hypoxia, continue to rise worldwide and (3) that AT inflammation is increasingly recognized as one of the initiating factors in the metabolic cascade leading to obesity-related complications.

### 4.4. Study Limitations

Some limitations and strengths of this study warrants discussion. First, the use of a single POP precludes extrapolation of our results to the more complex mixture of POPs preadipocytes/adipocytes are exposed to during their lifetime. Also, the single concentration of PCB126 (10 µM) used to maximize its effect on preadipocytes differentiation, despite being well tolerated by hSA, is likely falling off the physiologically range of total intracellular PCB levels observed in human AT, as reported by Bourez et al. [62]. Another limitation lies in the use of a single fixed hypoxia condition consisting of 3% O_2_. Despite the fact that 3% O_2_ is well recognized as being hypoxic in AT [15], this organ is likely exposed to fluctuations in O_2_ tensions of varying magnitudes, which may exert different effects on adipocyte metabolism. Hence, caution is warranted when extrapolating findings from our in vitro study to in vivo conditions.

## 5. Conclusions

The present study suggests that human subcutaneous preadipocytes exposed to PCB126 differentiate normally into adipocytes, but the resulting adipocyte population is characterized by lower ATP content, absolute glucose uptake and leptin expression. Exposure to hypoxia, on the other hand, exerts a much greater repression on adipocyte differentiation and lipid storage compared to PCB126 exposure during proliferation. Finally, our observations suggest that hypoxia and early PCB126 exposure have additive effects in terms of inflammatory response, a key feature of the development of chronic diseases such type 2 diabetes and cardiovascular diseases.

## Figures and Tables

**Figure 1 cells-12-02326-f001:**
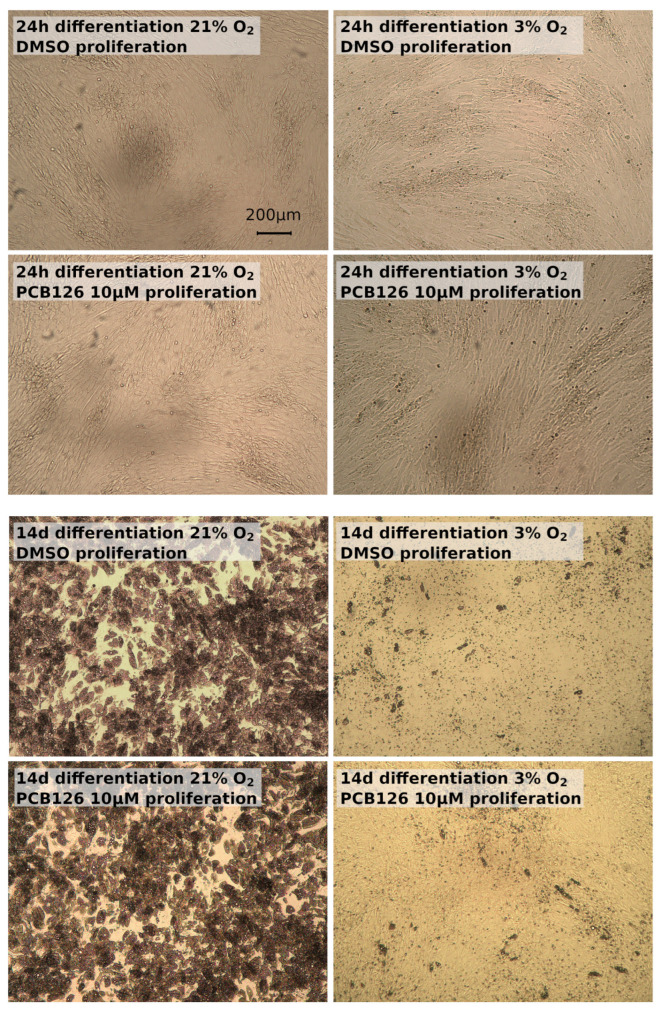
Differentiating preadipocytes exposed to either DMSO (vector) or 10 µM PCB126 during proliferation (3 days). The top 4 pictures were taken after the first 24 h of differentiation under either 21% or 3% O_2_. Bottom 4 pictures were taken after 14 days of differentiation under either 21% or 3% O_2_. All pictures were taken using a light microscope at 5× magnification.

**Figure 2 cells-12-02326-f002:**
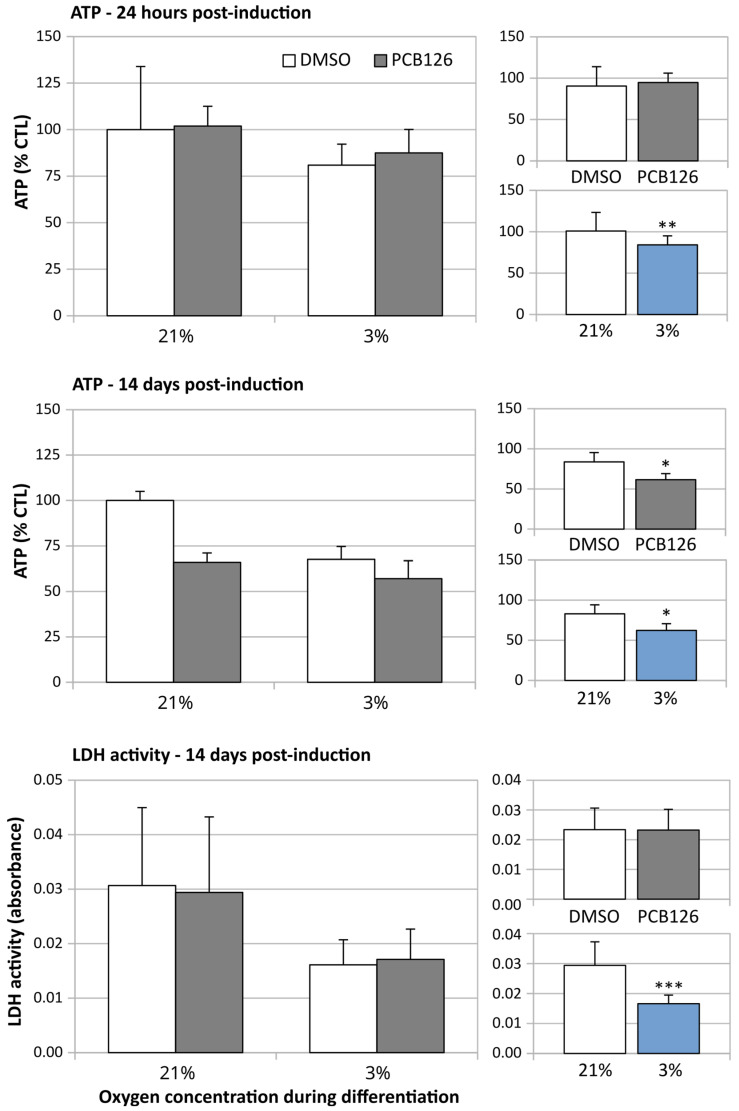
Effect of 10 µM PCB126 and hypoxia on cellular ATP content and media LDH activity. Top panels: Cell ATP content relative to control condition (DMSO, normoxia) 24 h following induction. Mid panels: Cell ATP content relative to control condition after 14 days of differentiation. Bottom panels: Media LDH activity measured after 14 days of differentiation. Main panels (left-hand side) illustrate the complete model and minor panels on the right-hand side summarize the separate main effects of PCB126 (upper minor panel) and hypoxia (lower minor panel). Main effect of PCB126 or hypoxia at * *p* < 0.05, ** *p* < 0.01 or *** *p* < 0.001. Results are expressed as mean ± SEM of three separate experiments for each treatment group.

**Figure 3 cells-12-02326-f003:**
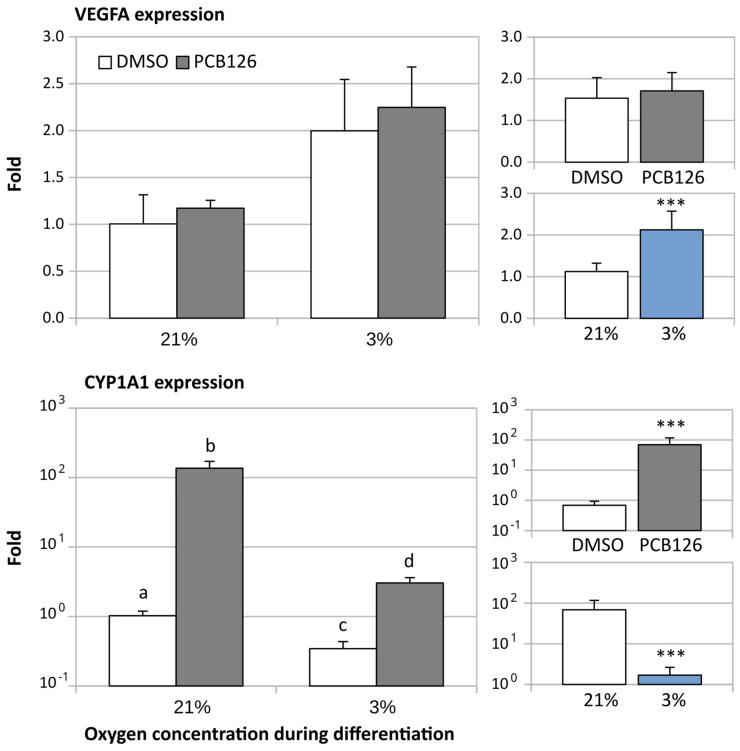
Effect of PCB126 and hypoxia on VEGFA (top panels) and CYP1A1 (bottom panels) expression in human adipocytes. Preadipocytes were treated with DMSO (vector) or 10 µM PCB126 for 3 days (pre-differentiation exposure), then PCB126 was removed from media and differentiation was induced. At this point, cells were subjected to 21 or 3% O_2_ for 14 days. VEGFA and CYP1A1 gene expression was measured using RT-qPCR. Main panels (left-hand side) illustrate the complete model and minor panels on the right-hand side summarize the separate main effects of PCB126 (upper minor panel) and hypoxia (lower minor panel). Bars not sharing a common letter are statistically different at *p* < 0.001. Main effect of PCB126 or hypoxia at *** *p* < 0.001. Results are expressed as mean ± SEM of three separate experiments for each treatment group.

**Figure 4 cells-12-02326-f004:**
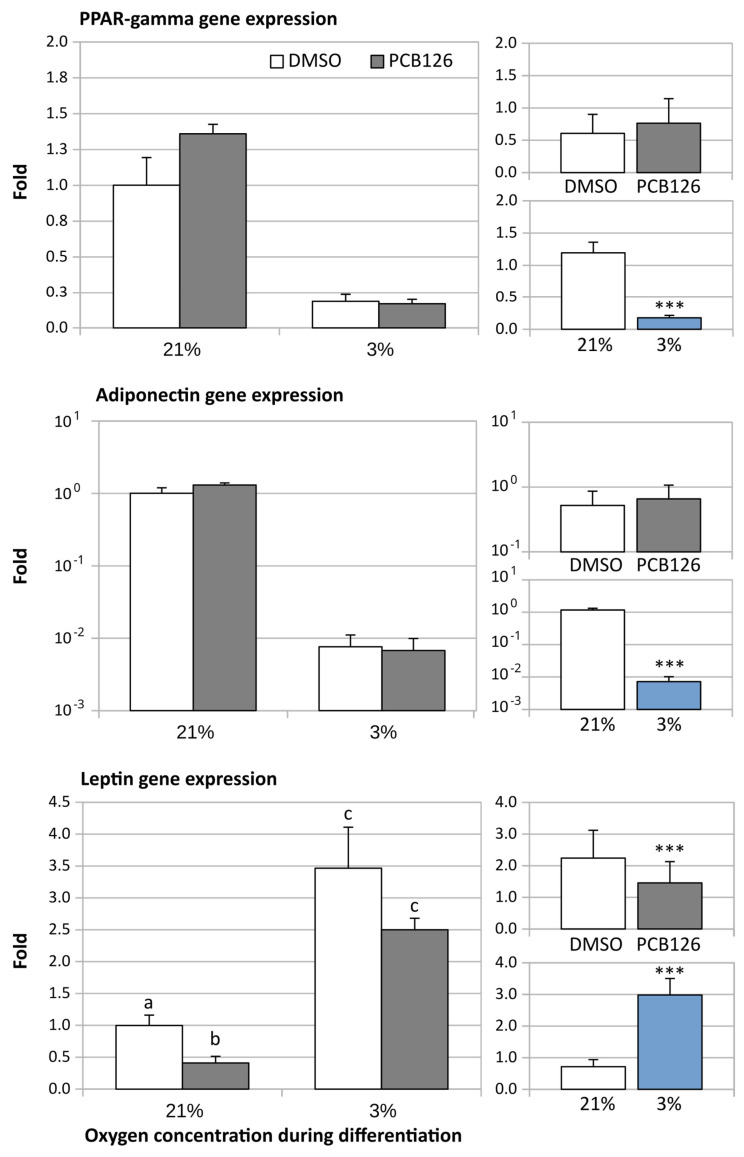
Effect of PCB126 and hypoxia on PPARγ (top panels), adiponectin (middle panels) and leptin (bottom panels) expression in human adipocytes. Preadipocytes were treated with DMSO (vector) or 10 µM PCB126 for 3 days (pre-differentiation exposure), then PCB126 was removed from media and differentiation was induced. At this point, cells were subjected to 21 or 3% O_2_ for 14 days. Gene expression of PPARγ, adiponectin and leptin was measured using RT-qPCR. Main panels (left-hand side) illustrate the complete model and minor panels on the right-hand side summarize the separate main effects of PCB126 (upper minor panel) and hypoxia (lower minor panel). Bars not sharing a common letter are statistically different at *p* < 0.001. Main effect of PCB126 or hypoxia at *** *p* < 0.001. Results are expressed as mean ± SEM of three separate experiments for each treatment group.

**Figure 5 cells-12-02326-f005:**
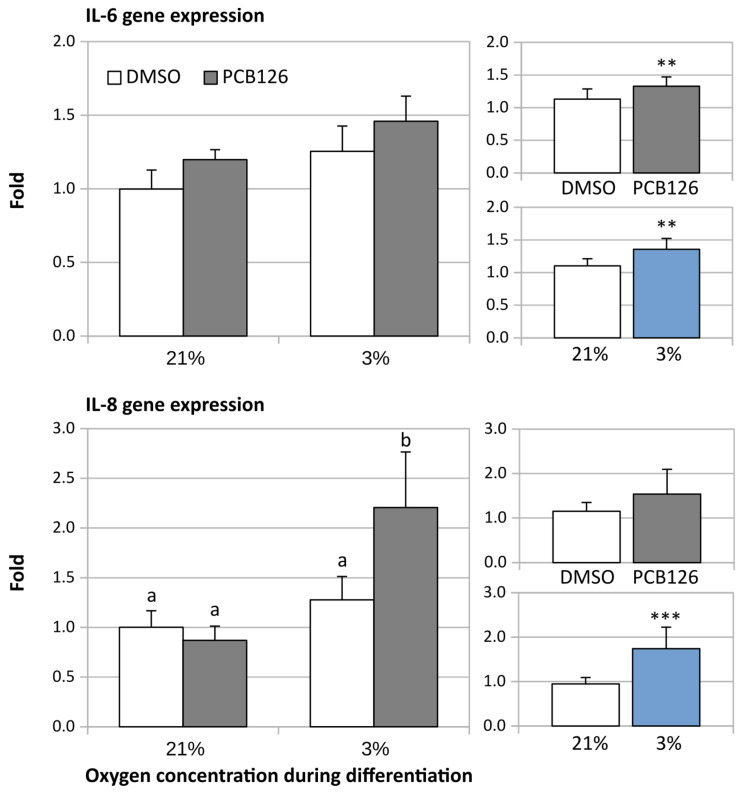
Effect of PCB126 and hypoxia on Il-6 (top panels) and IL-8 (bottom panels) expression in human adipocytes. Preadipocytes were treated with DMSO (vector) or 10 µM PCB126 for 3 days (pre-differentiation exposure), then PCB126 was removed from media and differentiation was induced. At this point, cells were subjected to 21 or 3% O_2_ for 14 days. Gene expression of IL-6 and IL-8 was measured using RT-qPCR. Main panels (left-hand side) illustrate the complete model and minor panels on the right-hand side summarize the separate main effects of PCB126 (upper minor panel) and hypoxia (lower minor panel). Bars not sharing a common letter are statistically different at *p* < 0.05. Main effect of PCB126 or hypoxia at ** *p* < 0.01 or *** *p* < 0.001. Results are expressed as mean ± SEM of three separate experiments for each treatment group.

**Figure 6 cells-12-02326-f006:**
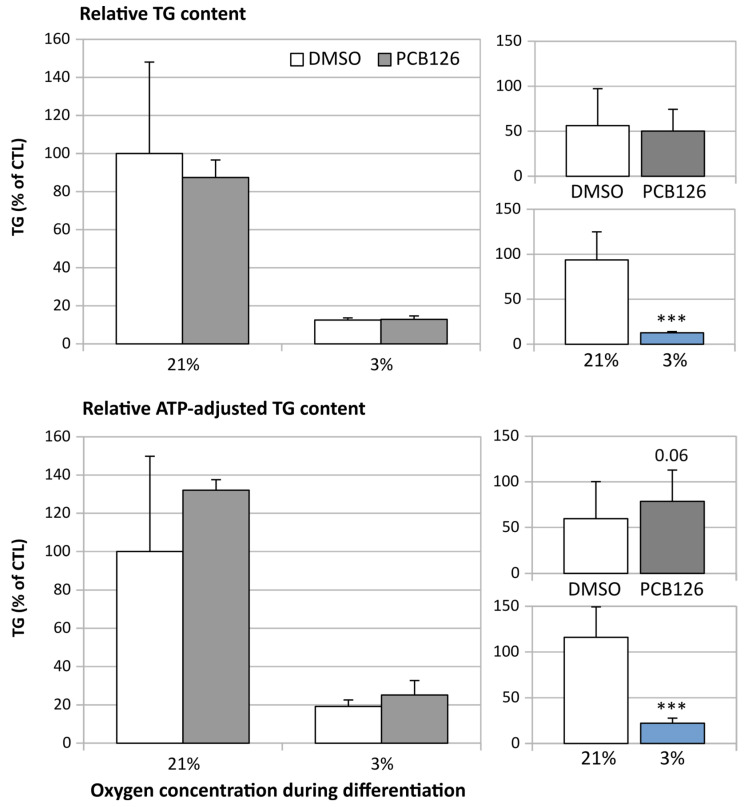
Relative TG content of human differentiated preadipocytes exposed to 10 µM PCB126 or DMSO for 3 days during proliferation followed by differentiation for 14 days under either 21% or 3% O_2_. Top panels illustrate relative TG content compared to DMSO/normoxia. Bottom panels illustrate TG content normalized to ATP, relative to DMSO/normoxia. Main panels (left-hand side) illustrate the complete model and minor panels on the right-hand side summarize the separate main effects of PCB126 (upper minor panel) and hypoxia (lower minor panel). Main effect of PCB126 or hypoxia at *** *p* < 0.001. Results are expressed as mean ± SEM of three separate experiments for each treatment group.

**Figure 7 cells-12-02326-f007:**
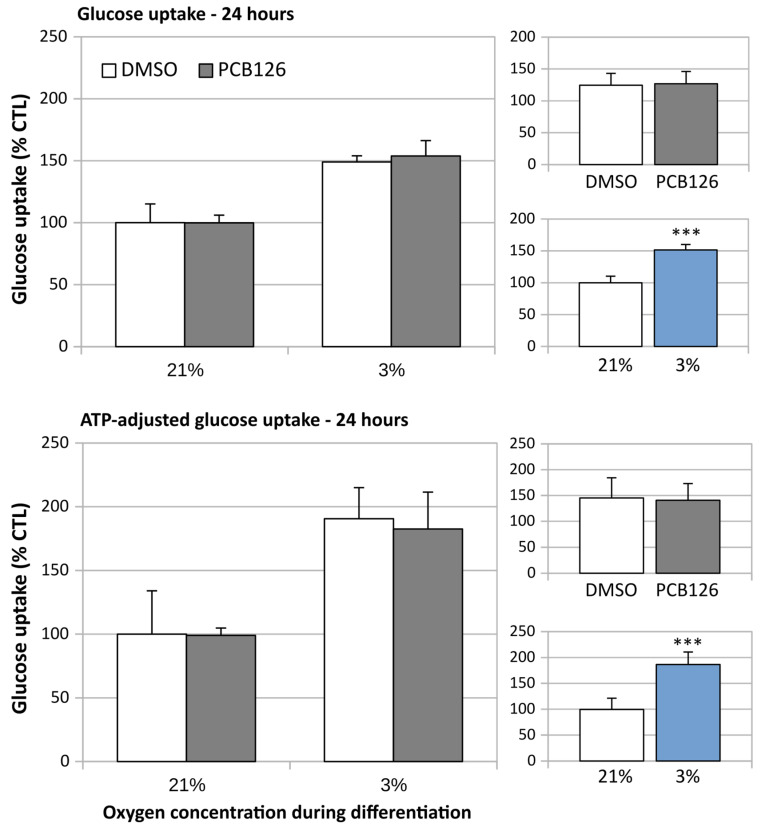
Early effect of PCB126 and hypoxia on basal glucose uptake in human adipocytes. Preadipocytes were treated with DMSO (vector) or 10 µM PCB126 for 3 days (pre-differentiation exposure), then PCB126 was removed from media and differentiation was induced. At this point, cells were subjected to 21 or 3% O_2_ for 24 h, after which glucose uptake was assessed. Main panels (left-hand side) illustrate the complete model and minor panels on the right-hand side summarize the separate main effects of PCB126 (upper minor panel) and hypoxia (lower minor panel). Main effect of PCB126 or hypoxia at *** *p* < 0.001. Results are expressed as mean ± SEM of three separate experiments for each treatment group.

**Figure 8 cells-12-02326-f008:**
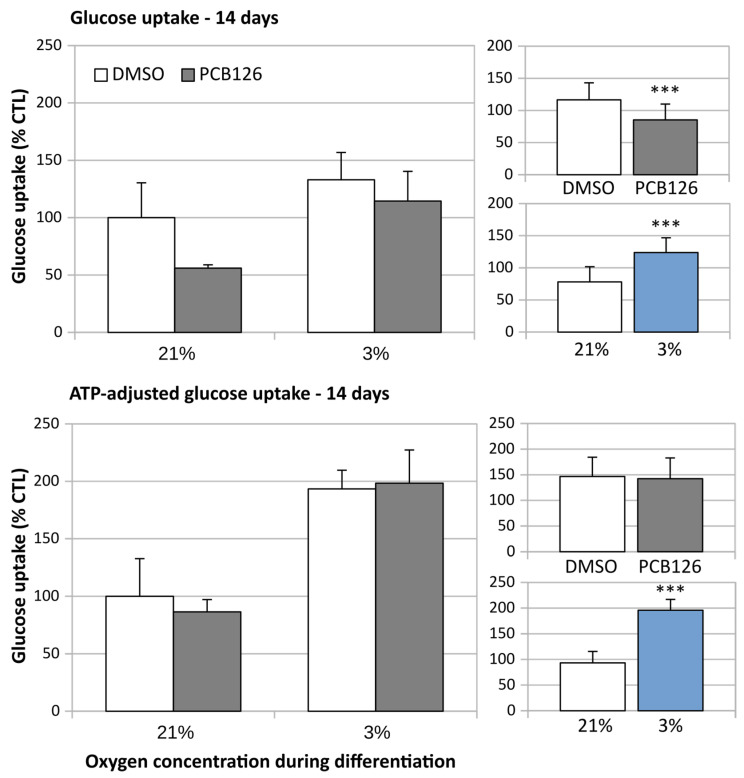
Effect of PCB126 and hypoxia on basal glucose uptake in human differentiated preadipocytes on day 14. Preadipocytes were treated with DMSO (vector) or 10 µM PCB126 for 3 days (pre-differentiation exposure), then PCB126 was removed from media and differentiation was induced. At this point, cells were subjected to 21 or 3% O_2_ for 14 days. Glucose uptake was assessed at the end of the 14-day differentiation period. Main panels (left-hand side) illustrate the complete model and minor panels on the right-hand side summarize the separate main effects of PCB126 (upper minor panel) and hypoxia (lower minor panel). Main effect of PCB126 or hypoxia at *** *p* < 0.001. Results are expressed as mean ± SEM of three separate experiments for each treatment group.

**Figure 9 cells-12-02326-f009:**
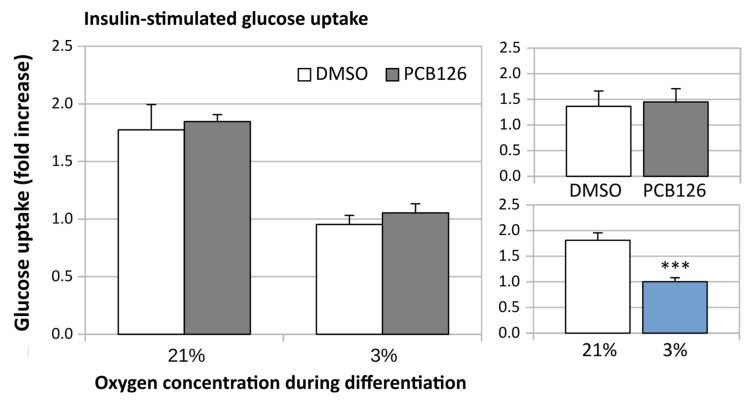
Effect of PCB126 and hypoxia on insulin-stimulated glucose uptake in human adipocytes. Preadipocytes were treated with DMSO (vector) or 10 µM PCB126 for 3 days (pre-differentiation exposure), then PCB126 was removed from media and differentiation was induced. At this point, cells were subjected to 21 or 3% O_2_ for 14 days and measurement was done on day 14 post induction. Main panels (left-hand side) illustrate the complete model and minor panels on the right-hand side summarize the separate main effects of PCB126 (upper minor panel) and hypoxia (lower minor panel). Main effect of PCB126 or hypoxia at *** *p* < 0.001. Results are expressed as mean ± SEM of three separate experiments for each treatment group.

**Figure 10 cells-12-02326-f010:**
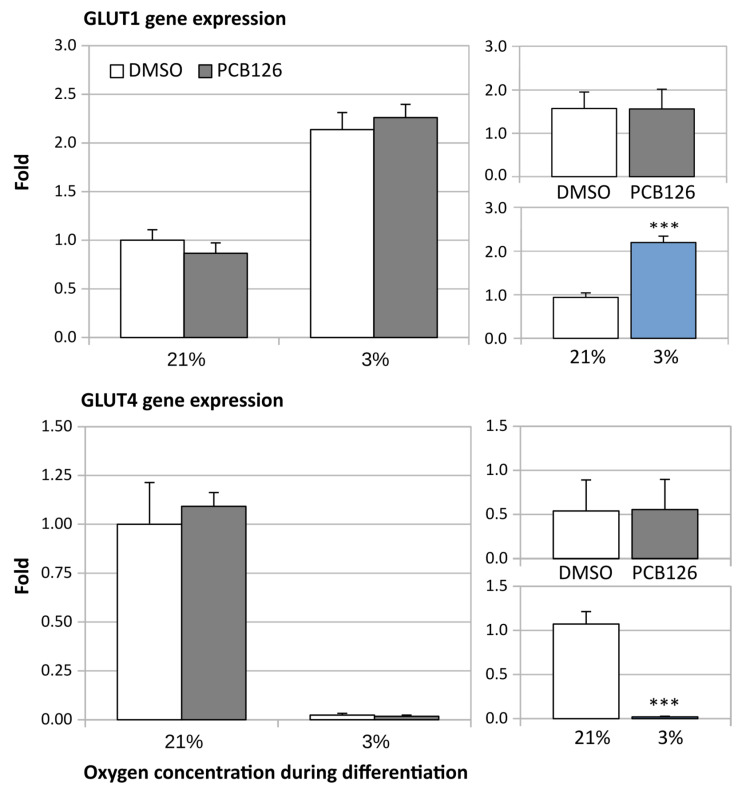
Effect of PCB126 and hypoxia on GLUT1 (top panel) and GLUT4 (bottom panel) expression in human adipocytes. Preadipocytes were treated with DMSO (vector) or 10 µM PCB126 for 3 days (pre-differentiation exposure), then PCB126 was removed from media and differentiation was induced. At this point, cells were subjected to 21 or 3% O_2_ for 14 days. Main panels (left-hand side) illustrate the complete model and minor panels on the right-hand side summarize the separate main effects of PCB126 (upper minor panel) and hypoxia (lower minor panel). Main effect of PCB126 or hypoxia at *** *p* < 0.001. Results are expressed as mean ± SEM of three separate experiments for each treatment group.

## Data Availability

Not applicable.

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
