# Peer review of "Human Preadipocytes Differentiated under Hypoxia following PCB126 Exposure during Proliferation: Effects on Differentiation, Glucose Uptake and Adipokine Profile"

_cells, 2023, doi:10.3390/cells12182326_

Round 1

Reviewer 1 Report

The study from El Amine et al. examined the human preadipocytes differentiated under hypoxia following PCB126 exposure during proliferation: effects on differentiation, glucose uptake and adipokine profile.

Here some comments related with this manuscript.

-In the abstract section, please add the concentration and exposure time of PCB126 in human subcutaneous adipocytes (hSA).

-The next phrase contains references repeated (page 2, line 45-46): “TCDD and dioxin-like PCBs have been shown to alter the acquisition of adipocyte phenotype (also called “adipocyte differentiation”) and adipocyte metabolism (9,10)(9,10).”

-Please define the acronyms ARNT, LDH, and PCDF.

-In methods section there are some words with different sizes (page 2, line 84-85).

-Please add the day of the experiment on which the preadipocytes were exposed to either 10uM PCB126 or DMSO, including the final percentage of DMSO.

-In the gene expression section, the amount of RNA total for Cdna synthesis should be indicated, including catalog number of genes analyzed.

-In the results section, in the microphotographs of figure 1 add scale bar.

-In my opinion, the histograms of most figures should have the statistical significance in the bars, and the legends of the figures should indicate the group comparisons because they are confusing. Please add in each legend figure if data are expressed as mean ± SD or SEM.

-In the figures 3 and 4, please indicate that the experiments are RT-qPCR.

-Why the expression of GLUT4 in insulin-stimulated human adipocytes was not evaluated (figure 9).

-The authors should evaluated some gene markers related to uptake and synthesis of triglycerides in the human preadipocytes differentiated under hypoxia following PCB126 exposure during proliferation.

-The authors should add a schematic diagram related to hypoxia and PCB126, considering the results obtained in differentiated preadipocytes.  

-There is repeated information on page 13, line 284-288. 

Author Response

Best regards!

Reviewer 2 Report

The authors describe a set of observational studies investigating the individual and combined effects of the dioxin-like chemical pollutant, PCB126, and hypoxia on human subcutaneous adipocytes. The novelty in this set of studies is the combination of PCB126 exposure alongside hypoxia and the treatment of human preadipocytes with PCB126 prior to differentiation. Hypoxia and dioxins share a common signaling pathway, which supports studying the combination of these treatments in adipocytes. The results largely indicate that hypoxia reduces adipocyte differentiation, increases inflammatory gene expression, and alters glucose uptake, with PCB126 treatment having a less robust impact on the outcomes measured.

·         The authors should comment on the appropriateness of using 10 µM PCB126 treatment. While the toxin accumulates in adipose tissue, it is not clear that 10 µM is physiologically relevant, and it appears to be at the higher end of the range of concentrations of PCB126 used in the literature.

·         The letter “u” is used often throughout the paper, including in figure legends, instead of the appropriate symbol for “micro”.

·         The significance of/rationale for the LDH measurement should be explained, and the abbreviation “LDH” should be explained.

·         Line 172-174 “…no obvious anomaly such as reduced cell proliferation or cell detachment from culture ware in response to treatments throughout the experiment duration (17 days total) (Figure 1)”. However, it appears that the hypoxia did have some effect on the cells in Fig 1, and the CellTiterGlo 2 assay suggests reduced cell viability. Please clarify.

·         While the CellTiterGlo 2 assay measures ATP, it is a measurement of cell viability, with reduced ATP concentrations indicating fewer viable cells. This is the standard use of the CellTiterGlo 2 kit that was used in these studies. Would the 20% reduction in ATP in response to hypoxia indicate reduced cell viability? The discussion seems to dismiss this possibility by suggesting the reduced ATP is due to hypometabolism and not reduced cell viability. The authors should conduct another cell viability assay that uses a different method than ATP quantification to determine if the treatments impact cell viability.

·         Line 179: a p value of <0.001 is cited, but according to the figure, it should read “p<0.01”.

·         Please indicate if standard deviation or standard error is used.

·         There are times that fold-change is used in the graphs, but the control group is not set to “1” precisely.

·         Line 494: the authors should refer to “leptin expression” instead of “leptin secretion”.

The manuscript needs to be edited for grammar and syntax. 

Author Response

Best regards!

Round 2

Reviewer 1 Report

I have no comments.